# Radiomics in Oncology, Part 2: Thoracic, Genito-Urinary, Breast, Neurological, Hematologic and Musculoskeletal Applications

**DOI:** 10.3390/cancers13112681

**Published:** 2021-05-29

**Authors:** Damiano Caruso, Michela Polici, Marta Zerunian, Francesco Pucciarelli, Gisella Guido, Tiziano Polidori, Federica Landolfi, Matteo Nicolai, Elena Lucertini, Mariarita Tarallo, Benedetta Bracci, Ilaria Nacci, Carlotta Rucci, Marwen Eid, Elsa Iannicelli, Andrea Laghi

**Affiliations:** 1Radiology Unit, Department of Medical Surgical Sciences and Translational Medicine, Sapienza University of Rome-Sant’Andrea University Hospital, Via di Grottarossa, 1035-1039, 00189 Rome, Italy; damiano.caruso@uniroma1.it (D.C.); michela.polici@uniroma1.it (M.P.); marta.zerunian@uniroma1.it (M.Z.); francesco.pucciarelli@uniroma1.it (F.P.); gisella.guido@uniroma1.it (G.G.); tiziano.polidori@uniroma1.it (T.P.); federica.l2005@libero.it (F.L.); matteo.nicolai@uniroma1.it (M.N.); elena.lucertini@uniroma1.it (E.L.); benedetta.bracci@uniroma1.it (B.B.); ilaria.nacci@uniroma1.it (I.N.); carlotta.rucci@uniroma1.it (C.R.); elsa.iannicelli@uniroma1.it (E.I.); 2Department of Surgery “Pietro Valdoni”, Sapienza University of Rome, 00161 Rome, Italy; mariarita.tarallo@uniroma1.it; 3Internal Medicine, Northwell Health Staten Island University Hospital, Staten Island, New York, NY 10305, USA; eid.marwen@gmail.com

**Keywords:** Radiomics, Oncologic Imaging, Radiomics technical principles

## Abstract

**Simple Summary:**

This Part II is an overview of the main applications of Radiomics in oncologic imaging with a focus on diagnosis, prognosis prediction and assessment of response to therapy in thoracic, genito-urinary, breast, neurologic, hematologic and musculoskeletal oncology. In this part II we describe the radiomic applications, limitations and future perspectives for each pre-eminent tumor. In the future, Radiomics could have a pivotal role in management of cancer patients as an imaging tool to support clinicians in decision making process. However, further investigations need to obtain some stable results and to standardize radiomic analysis (i.e., image acquisitions, segmentation and model building) in clinical routine.

**Abstract:**

Radiomics has the potential to play a pivotal role in oncological translational imaging, particularly in cancer detection, prognosis prediction and response to therapy evaluation. To date, several studies established Radiomics as a useful tool in oncologic imaging, able to support clinicians in practicing evidence-based medicine, uniquely tailored to each patient and tumor. Mineable data, extracted from medical images could be combined with clinical and survival parameters to develop models useful for the clinicians in cancer patients’ assessment. As such, adding Radiomics to traditional subjective imaging may provide a quantitative and extensive cancer evaluation reflecting histologic architecture. In this Part II, we present an overview of radiomic applications in thoracic, genito-urinary, breast, neurological, hematologic and musculoskeletal oncologic applications.

## 1. Introduction

Radiomics is an emerging tool used in oncologic imaging with the future perspectives to become central in the workup of cancer patients. This imaging technique was extensively investigated in tumor diagnosis, prognosis evaluation and response to therapy. Radiomics also provides an estimation of delta tumor heterogeneity and aggressiveness, before and after cancer therapy. It can be used as a bridge between histology and medical images, useful for clinicians to manage patients with a patient tailored workflow [1,2,3]. Additionally, quantitative analysis of medical images might provide additional evidence to the traditional tissue biopsy which is often affected by sample bias [4]. The lack of standardization limits the routine use of Radiomics as a clinical tool, although the increasing number of encouraging results supports the use of Radiomics in cancer patient management [5,6].

Technical principles and applications of Radiomics to gastrointestinal tumors were included in the Part 1 of this review, with the aim to provide an organic explanation of techinical aspects and limitations as the first issue, then start to discuss Radiomics in the gastrointestinal cancers, representing one of the main application field in terms of types of malignancy.

The objective of part 2 of this paper is to review the main oncologic radiomic studies centered on diagnosis, prognosis and response to therapy in thoracic, genito-urinary, breast, neurological, hematologic and musculoskeletal malignancies.

## 2. Lung Cancer

Recently, Radiomics has been extensively applied on lung cancer and multiple studies evaluated its role in diagnosis, prognosis and response to treatment. Table 1 summarizes the main studies discussed about the application of Radiomics on lung cancer diagnosis, prognosis and response to treatment.

Regarding lung cancer diagnosis, Radiomics was used to distinguishing non-small cell lung cancer (NSCLC) from pulmonary granulomas and small cell carcinoma (SCLC) from adenocarcinoma. Radiomics achieved strong results in the initial diagnosis, with good differentiation of lung neoplasms, with histology as a reference standard [7,13,14]. In particular, Radiomics has been tested in distinguishing lung adenocarcinoma from granulomas by Beig N. et al. [7] in 290 patients who were retrospectively analyzed in a multicenter study. Nodule shape, wavelet and Haralick texture features were extracted from intra- and peri-nodular regions of interest (ROIs). Radiomic models were developed and trained by using machine learning classifiers and these were compared against Convolutional Neural Network (CNN) approach and visual nodule assessment by two radiologists. Results showed higher performance of the combined intra-nodular and peri-nodular with radiomic features (AUC 0.80) compared with CNN (AUC 0.76) and radiologist assessments (AUC 0.61 and 0.60). Similar results were shown by Linning E. et al. [8], who applied Radiomics in the binary discrimination between SCLC and adenocarcinoma on unenhanced CT (AUC 0.822), and in distinguishing adenocarcinoma from squamous cell carcinoma (AUC 0.655). Furthermore, Radiomics showed good accuracy in the preoperative discrimination of malignancy behavior of ground-glass nodules resulting in a better independent predictor over CT morphology or mean CT value [15].

In prognosis prediction and pre therapeutic assessment, Radiomics has been tested to stratify patients between low- and high-risk, in relation to overall survival and having a mutational panel preoperatively. Results showed Radiomics to have a reasonable accuracy, especially when combined with clinical data [9,10,16,17]. Cong M. et al. [9] investigated metastatic lymph-node prediction before surgery in stage IA NSCLC. They demonstrated that radiomic model alone, extracted from 649 primary lung lesions identified on baseline contrast enhanced CT scan, reached a diagnostic accuracy of 0.78 (AUC 0.898) and 0.80 (AUC 0.851) for test and validation cohorts, respectively. When using a combined model (radiomic and clinical ones) they achieved higher diagnostic accuracy of 0.80 (AUC 0.911) and 0.83 (AUC 0.86) for test and validation cohorts, respectively. Radiomics also showed interesting results in positron emission tomography (PET). Zhang J. et al. [10] demonstrated that Radiomics alone and combination models significantly performed better than clinical models alone in discriminating epidermal growth factor receptor (EGFR) mutation status in NSCLC. The possibility to derive genetic data with a non-invasive way would overcome the limitation of lung biopsy and allow for a more precise treatment with tyrosine kinase inhibitors (TKIs). Nardone V. et al. also showed the potential of Radiomics in identifying NSCLC patients who may benefit from anti-PD-1 antibody treatment (i.e., Nivolumab). Moreover, by performing highly reproducible texture parameters’ cut-off, Radiomics was able to discriminate low from high risk patients with the aim to select patients for Nivolumab therapy [17].

Regarding treatment assessment and response to therapy, Radiomics reached reasonable performances in correlation with clinical outcomes [11,12]. Zerunian M. et al. [11] investigated the application of Radiomics on outcomes prediction in patients treated with immunotherapy, in particular first line Pembrolizumab. Texture features extracted from CT images were significantly associated with lower overall survival (OS) and progression-free survival (PFS) (*p* < 0.0035, AUC of 0.72 and 0.74, respectively). Khorrami M. et al. [12] showed that derived intra-tumoral and peri-tumoral CT texture shape features reached an AUC of 0.90 and 0.86 in the training and test set in prediction of major response after neoadjuvant radio-chemotherapy in stage IIIA NSCLC.

## 3. Uterine Cancer

Recently, Radiomics has been investigated in order to improve diagnostic accuracy, assessment staging, prognosis prediction and response to therapy both in uterine cervix and uterine corpus cancer.

In the initial staging and prognosis prediction, Radiomics was tested as a potential tool to assess metastatic nodes in the pre-operative setting with promising, which is important information for the clinicians as they attempt to tailor therapeutic approaches to each patient and to predict patient prognosis [18,19]. De Bernardi et al. [18] tested Radiomics as a predicting nodal metastases tool by analyzing 115 primary endometrial cancer on preoperative ^18^F-FDG-PET/CT. They showed that only one heterogeneity feature (GLSZM ZP) was able to increase sensitivity (75–89%) to predict nodal metastases in comparison with visual assessment (33–50%), nevertheless the specificity was higher by using visual detection (95–99% vs. 80–81%). In addition to radiomic features extracted from the primary tumor, other authors have focused in evaluating measurements from lymph nodes [19]. Kan Y. et al. [19] investigated if radiomic signature multiple-sequence MRI could be used as a noninvasive biomarker for preoperative lymph nodes (LN), reflecting tumor aggressiveness and influencing patient staging, prognosis and therapeutic approach in cervical cancer. They retrospectively built a radiomic signature on 143 baseline MRIs (T1w contrast-enhanced (CE) and T2w), divided into test and validation cohorts. After 3D manual segmentation of lymph-nodes, radiomic parameters were extracted and diagnostic accuracy was evaluated using histology reports as reference standard. Results underlined that morphological MRI nodal evaluation is not satisfactory for predicting nodal metastases (0.65 and 0.49, in test and validation cohorts, respectively), while radiomic signature allowed for improved accuracy to discriminate LN with an accuracy of 0.75 and 0.72 in test and validation cohorts, respectively.

Another relevant radiomic application is its use as an imaging biomarker able to predict response to chemotherapy by analyzing tumoral and peritumoral tissues and by combining outcome data. Overall, Radiomics reached higher accuracy and performance in comparison with clinical models alone [20,21,22]. Sun C. et al. [20] investigated if radiomic features, based on pre-treatment MRIs, could predict clinical response to neoadjuvant chemotherapy in patients with locally advanced cervical cancer. A multicentric retrospective study was performed on 275 patients and different radiomic models were tested by using several combinations of radiomic features, extracted from intra-tumoral and peritumoral tissues from MRIs on T1w and T2w sequences; the performance of the radiomic model was analyzed and compared to a clinical model (FIGO stage, age and gross type). The radiomic model, including all the sequences, showed a 30% higher performance in treatment response prediction after NAC in comparison with the clinical model alone (AUC 0.998 and 0.999 vs. 0.666 and 0.608 for training and testing cohorts, respectively). Promising results in DFS prediction were also demonstrated by Lucia F. et al. [21], who validated, in a multicenter study, a radiomic model using features extracted by ^18^F-FDG PET/CT and ADC map on MRI. The combined model reached an accuracy with a range between 96% and 98%, higher than clinical models available with an accuracy of 56–60%. Radiomic models have also been assessed as a biomarker to predict response to radiotherapy by Takada A. et al. [22]; they showed that including peri-tumoral tissue in the volume of interest resulted in higher accuracy in the prediction of recurrence.

Regarding uterine corpus tumors, Radiomics was mainly explored, with promising results in the accuracy, risk stratification and pre-treatment prognosis of endometrial cancer and differential diagnosis in uterine sarcoma. Particularly, Radiomics was investigated as an imaging tool in the planning of therapeutic approaches that are patient-centered according to nodal involvement status and endometrial cancer or sarcoma aggressiveness [23,24,25]. In the evaluation of pelvic LN as a predictor of patient prognosis, Yan B.C. et al. [23] used a random forest classifier to build an MRI radiomic model based on segmentation of primary endometrial cancer. The diagnostic performance and clinical net benefits were compared between MRI alone and MRI Radiomics-aided, with higher diagnostic performance reached by using the radiomic model. The AUC, CDC (clinical decisive curve), NRI (net reclassification index) and IDI (discrimination index) were indeed higher for the Radiomics-aided model, than for MRI alone. For surgery planning, Yan B.C. et al. [24] also assessed high-risk endometrial cancer, using Radiomics on MRI (T2WI, DWI, ADC and CE-T1WI sequences) and developed a nomogram. The model obtained by combining radiomic and clinical parameters showed good performance and proved the usefulness of Radiomics for surgical management of endometrial cancer. MRI and radiomic features were also analyzed by Xie H. et al. [25], for differentiating uterine sarcoma from atypical leiomyoma. The radiomic model was based on Apparent Diffusion Coefficient (ADC) maps and was compared with diagnostic efficacy of experienced radiologists; AUCs showed comparable diagnostic efficacy in differentiating uterine sarcoma from atypical leiomyoma.

## 4. Ovarian Cancer

Accurate diagnosis of ovarian masses is a field of interest for Radiomics, as it is a potential non-invasive biomarker to characterize adnexal lesions. Table 2 summarizes the main studies and radiomic results applied to ovarian masses. Radiomic analysis yielded interesting results in ovarian masses differential diagnosis, both in identifying histological type and in distinguishing among benign, borderline and malignant masses, compared to conventional imaging [26,27]. Figure 1 provide a graphic representation of radiomic workflow from image segmentation to statistical analysis. An interesting retrospective study was performed by Zhang H. et al. [26], who manually drew ROIs on T1w, T2w and ADC maps of 286 patients before adnexal surgery. They extracted and selected radiomic features with the least absolute shrinkage and selection operator method (LASSO) and tested the radiomic model’s ability to differentiate benign from malignant adnexal tumors and to identify type I and type II ovarian epithelial cancer (OEC). Results showed higher accuracy of Radiomics, compared with traditional radiologist evaluation, in distinguishing malignant adnexal tumors from benign masses (90% vs. 83.5% for leave-one-out cross-validation cohort and independent validation cohort); an accuracy of 83% (AUC 0.85) was reached in the discrimination of type I and type II OEC. The most relevant radiomic features identified were low-high-high, short-run high gray-level emphasis extracted from coronal T2w for benign and malignant tumor discrimination, and low-low-high variance from coronal T2w images to differentiate type I from type II OEC. Another feasibility study was performed by Song X.L. et al. [27] testing Radiomics efficiency, derived from CE-MRI pharmacokinetic (PK) protocol, in discriminating among benign, borderline and malignant adnexal neoplasms by using 2- and 3-class classification predictive tasks. The analysis retrospectively assessed 104 CE-MRI divided into training and validation cohorts. Two-class classification task results showed an AUC of 0.899, 0.865 and 0.893 for the following discrimination tasks, respectively: benign vs. borderline, benign vs. malignant and borderline vs. malignant. ROCs for 3-class classification (one single ovarian tumors vs. other) showed good diagnostic performances with AUC of 0.893, 0.944 and 0.891 for the benign, borderline and malignant tumors, respectively.

In regards to prognosis, several studies performed showed significant correlation between radiomic and outcome data, but not significant results identifying the BRCA mutation panel was found [28,29]. Meier A. et al. [28] investigated the hypothesis that Radiomics could evaluate loco-regional and distant site tumor heterogeneity of high-grade serous ovarian cancer and link it to BRCA mutation status and outcome prediction. The retrospective study was performed on 88 preoperative CT scans and a slice-by-slice segmentation was performed both on primary ovarian masses and distant metastatic implants. Haralick texture features, after computing a gray-level co-occurrence matrix, were extracted and inter-site texture heterogeneity was tested including inter-site entropy, cluster variance and cluster prominence. Results showed that higher values of textural features significantly correlated with lower OS and PFS. Additionally, BRCA-negative patients showed to have a significant association among all high values of selected metrics and lower complete surgical resection. No significant results were reached for all the three metrics to distinguish between BRCA mutation/-non mutation patients. Another study, performed on a larger population by Lu H. et al. [29] analyzed a radiomic profile containing 657 features for 364 EOC cases. They built a radiomic prognostic vector based on 4 descriptors derived from the primary tumor able to identify patients with median OS less than 2 years.

In response to therapy, Radiomics showed some promising results especially when combined with radiological disease burden and clinical data. In addition, the calculation of delta radiomic features between baseline and post-chemotherapy CT scans reached positive results in the prediction of tumor response [30,31]. Himoto Y. et al. [30] analyzed 75 patients with recurrent EOC, treated with immunotherapy, on contrast-enhanced CT. Radiomic analysis, including inter and intra-tumor heterogeneity features, were performed and combined with radiological disease burden and clinical data. Univariate analysis showed that reduce disease sites and both lower inter- and intra-tumor heterogeneity were significantly associated with durable clinical benefit. Higher cluster-site dissimilarity represented an independent indicator of shorter time to off-treatment with a Hazard Ratio of 1.19. Multivariate analysis confirmed that higher energy of the larger lesion, as indicator of lower intratumor heterogeneity and fewer disease sites, was a predictor of durable clinical benefit. Similarly, Danala G. et al. [31] investigated the role of quantitative analysis in early prediction of cancer response to chemotherapy by using imaging biomarkers extracted from pre- and post-therapy 91 CT scans. In particular, they calculated delta features and reached positive results (AUC 0.771), with the strongest performance (AUC 0.81 and 0.829) achieved using the fusion models, combining the optimal features derived from pre-treatment CT scan and delta features.

## 5. Prostate Cancer

Another area where Radiomics may be a promising tool is in non-invasive prostate cancer (PCa) assessment, by providing a correct detection, PI-RADS V2.1 correlation, risk-stratification and radiotherapy planning [32,33]. Figure 2 provides a graphic representation of radiomic analysis in prostate cancer, from image segmentation to signature validation.

Regarding PCa detection, Radiomics was investigated as a pre-operative imaging tool able to detect highly suspected malignant prostate nodules in order to identify optimal biopsy sites. In that regard, quantitative approaches yielded superior results in comparison with traditional imaging assessment [34,35,36]. Lay N. et al. [34] tested Radiomics as a tool to recommend biopsy sites, by combining spatial, intensity and texture features with random forest classification on 224 patients by analyzing T2w, ADC and Diffusion Weighted Imaging (DWI) at b2000 sequences. The authors reported better performance of this method (AUC 0.93) compared with previous studies testing support vector machine (SVM) on the same data (AUC 0.86). Cuocolo R. et al. [35] studied shape features in distinguishing clinically significant (41 patients) and non-clinically significant (34 patients) prostate lesions aiming at reducing the false positive rate seen with MRI. In particular, ten shape features were extracted from multiparametric MRI (mpMRI) and the univariate analysis reported that almost every shape feature was statistically significant between clinically significant and non-clinically significant groups, whereas the multivariate analysis showed that only radiomic surface area to volume ratio, extracted from ADC maps, was an independent predictor of PCa. Similarly, Wibmer A. et al. [36] showed promising results of Radiomics in PCa detection and Gleason Score (GS) assessment on mpMRI, in particular Gleason Score (GS) was significantly associated with higher values of Entropy (GS 6 vs. 7: *p* = 0.0225; 6 vs. >7: *p* = 0.0069) and lower values of energy (GS 6 vs. 7: *p* = 0.0116, 6 vs. >7: *p* = 0.0039) derived from ADC maps. Furthermore, Qi Y. et al. [37] created a radiomic model by using a random forest classifier, based on 2104 features extracted from MRI sequences. The combined model (radiological and clinical data) returned AUC values of 0.956 and 0.933 on the test (*n* = 133) and validation (*n* = 66) population, respectively, making it an additional potential tool for the clinicians in treatment decision-making.

To date, risk-assessment of PCa recurrence is based on clinical parameters (i.e., GS, PSA level, cancer grading and tumor stage) and no objective and accurate tools to stratify cancer patient into low- and intermediate-risk is currently available. In that context, Radiomics is a promising tool to support clinical management of these patients and achieved good results in stratifying patients according to risk of recurrence [38,39,40]. Recently, Gugliandolo S.G. et al. [38] obtained a radiomic signature, from 65 mpMRI (T2w images) of localized PCa, to distinguish low- from intermediate-risk patients. Texture features were the main predictive parameters of Gleason Score, PI-RADS and risk-classification, while intensity domain was strictly linked to T-stage, extracapsular extension score and risk-classification (AUC ranging from 0.74 to 0.94). Similarly, Osman S.O.S. et al. [39] used Radiomics to assess Gleason Score and risk-assessment in 342 PCa patients. CT-based Radiomics yielded excellent results in distinguishing low- from intermediate-risk (AUC 1.00) and low- from high-risk (AUC 0.96). Furthermore, Algohary A. et al. [40] proposed MRI-based radiomic signature (T2w and DWI) to assess the presence of significant disease in patients in active surveillance who had a discordance between the histopathological and PI-RADS findings. They obtained promising in identifying significant disease with *p* < 0.001.

Radiomics was also tested to evaluate prognosis in the high-risk patients who underwent radical prostatectomy. Bourbonne V. et al. [41] purposed to assess the prognostic value of radiomic signature MRI-based in planning for adjuvant radiotherapy. Radiomics showed strong results in predicting biochemical recurrence and was an independent prognostic factor of biochemical relapse free survival after radical prostatectomy.

## 6. Urinary System

Radiomics was extensively investigated in the workup of bladder cancer (BC) and kidney cancer, especially in the assessment of tumor grading and local invasion, and only few studies tested Radiomics in the prediction of response to therapy. Several studies support the use of Radiomics in the assessment of cancer grading and local invasion both in bladder and kidney cancer [42,43,44]. Zhang G. et al. [42] developed a CT-radiomic model to assess bladder cancer grading, in order to distinguish low- from high-grade cancer. They analysed 145 lesions (108 of training and 37 of validation datasets) which were manually segmented on CT-urography. They obtained an AUC of 0.95 and 0.86 in training and validation dataset, respectively. Similarly, Goyal A. et al. [43] tested texture analysis on MRI to assess renal cell carcinoma (RCC) grading and demonstrated that several parameters, such as entropy, mean positive pixels and mean reached promising AUCs (0.823, 0.870 and 0.889, respectively) in differentiating RCC with low- and high-grade.

In the assessment of local invasion of bladder cancer, Radiomics showed promising results in the evaluation of muscular invasion [45,46]. Xu S. et al. [45] built a radiomic model combining radiological and clinical data based on MRI (DWI) and transurethral resection, that yielded better results in muscular invasion assessment (sensitivity 0.964, accuracy 0.897) in comparison with transurethral resection alone (sensitivity 0.655) or MRI evaluation alone (sensitivity 0.764). Similarly, Zheng J. et al. [46] developed an MRI-based radiomic-clinical nomogram with the same objective and demonstrated an AUC of 0.922 in differentiating muscle from non-muscle invasion.

In relation to response to therapy, Radiomics was investigated both in bladder cancer and RCC yielding positive results in identifying good responders after neoadjuvant chemotherapy [47,48,49]. Cha K.H. and colleagues [47] tested CT-based deep learning CNN and Radiomics to identify bladder cancer patients with muscle-invasion who had a complete response to neoadjuvant chemotherapy. The authors demonstrated that quantitative methods could identify complete responders with an AUC of 0.80. Previously, only a few studies evaluated the use of CT based features to predict response to therapy in metastatic RCC. Particularly, Smith A.D. at al. [48] proposed to quantify the initial CT changes in cancer vascularity to predict the response to antiangiogenic therapy in metastatic RCC patients, and they obtained significant results in distinguishing responder from non-responder patients. Additionally, Goh V. et al. [49] tested the ability of CTTA after two cycles of therapy in metastatic RCC and it was shown to be a predictive imaging biomarker of response and cancer heterogeneity and possibly an independent parameter of progression.

## 7. Breast Cancer

Recently, Radiomics has been studied as a non-invasive imaging biomarker, combined to conventional radiology and breast biopsy, to overcome the major intrinsic limitations in breast cancer (BC) assessment in early diagnosis, tumor biology and response to therapy.

In differential diagnosis, Radiomics was used to distinguish benign from malignant lesions on MRI by analyzing tumoral and peritumoral tissues [50,51]. Zang Q. et al. [50] built a radiomic MRI-based model, by performing manual and volumetric segmentation, to differentiate benign from malignant breast nodules. This model was developed and validated on 95 benign and 112 malignant nodules (training and validation group with 159 and 48 nodules, respectively). All patients underwent MRI (T1w, T2w, DWI and DCE) and the model was tested on both morphological and functional sequences, alone and in combination, by using an SVM classifier. The combining model, including features extracted from T2w, diffusion kurtosis imaging (DKI), ADC map and DCE, reached an AUC of 0.921 and accuracy of 0.833. Similarly, Zhou J. et al. [51] tested radiomic and deep learning methods on DCE-MRI sequences, by performing 3D automatic, quantitative segmentation of tumoral and peritumoral tissues. The authors showed promising results using a deep-learning approach by achieving accuracy of 91%.

Concerning the identification of key prognostic factors, such as nodal involvement and HER-2 status, Radiomics was tested as a non-invasive imaging biomarker, combined with clinical data, reaching good performance [52,53,54]. Gao Y. et al. [52] tested a radiomic US-based nomogram in 343 patients affected by T1/T2 invasive BC by adding patient age and tumor size. The model showed good performance to detect nodal metastases in both the training and validation cohorts (0.846 and 0.733 AUC, respectively) by analyzing primary breast cancers. Liu C. et al. [53] developed an MRI-based radiomic model, combining quantitative features and clinical data, and reached a n AUC of 0.869 and a negative predictive value (NPV) of 0.886. Regarding the assessment of HER-2 status Zhou J. et al. [54] tested Radiomics on 306 BC mammography and a sensitivity, specificity, accuracy and AUC (87%, 59%, 80% and 0.85, respectively).

For BC response to therapy, radiomic analysis yielded pivotal results in recognizing responder patients, an essential component for optimal therapeutic management [55,56,57]. Braman N. et al. [55] tested Radiomics on MRI to assess response to HER2-target therapy, by extracting 209 textural features from primary and peritumor tissues. They showed a promising performance both in predicting HER2 positive patients (AUC 0.89) and target-therapy response (AUC 0.80). Liu Z. et al. [56] built and validated a radiomic model to predict pathological complete response to neoadjuvant therapy in HER2+, HER- and triple-negative patients. This model achieved good results, in both the training and validation datasets, in each different patient groups. Similarly, Antunovic L. et al. [57] showed the ability of Radiomics to predict complete response to neoadjuvant chemotherapy in locally advanced BC, by using features extracted from baseline FDG PET/CT, with quite good performance (AUC 0.70–0.73).

## 8. Neurological System

For decades, management of central nervous system tumors focused on conventional MRI and CT. However, Radiomics has been playing an increasingly important role as a non-invasive imaging biomarker and is increasingly investigated [58] especially in glioblastomas, the most aggressive and poor prognosis main brain cancer focusing grading, mutational status and response to therapy. The main radiomic studies in neuro-oncology were summarized on Table 3.

Radiomics was tested at diagnosis to define gliomas grading, particularly focusing on distinguishing low (Grade I and II) from high grade gliomas (Grade III and IV) [59,60]. Tian Q. et al. [59] analyzed texture analysis on MRI to distinguish low-grade gliomas (LGG) from high-grade gliomas (HGG) in 153 patients (42 LGG and 111 HGG) with SVM-based classification model. Excellent results were obtained both in comparison between of LGG and HGG (AUC of 0.987, accuracy 96.8%, sensitivity 96.4% and specificity 97.3%) and in distinguishing grade III from IV (AUC 0.992, accuracy 98.1%, sensitivity 98.7% and specificity 97.4%). Similarly, Cho H. H. et al. [60] compared LGG and HGG gliomas by performing a manual segmentation in a semi-automatic method and achieved an AUC of 0.903. The Authors demonstrated that gliomas grading could be estimated by using high-dimensional features with good accuracy.

In relation to prognosis assessment, Radiomics was found to correlate strictly with specific molecular patterns of gliomas [61,62]. Molecular evaluations are routinely assessed by tumor biopsy, which could be affected by sample bias, low repeatability and iatrogenic complications. In particular, Chang P. et al. [61] tested a deep-learning approach, enforced with CNN, on 259 MRI of gliomas and CNN was shown to have high accuracy in identifying isocitrate dehydrogenase mutations (94%), 1p/19q co-deletion (92%) and O(6)-Methylguanine-DNA methyltransferase (MGMT) promoter methylation (83%) status. Li Z.C. et al. [62] established the role of Radiomics in pretreatment prediction of MGMT promoter methylation status, with the aim to overcome the limitations of biopsy, obtaining a predictive model with promising accuracy (AUC 0.88, accuracy 80%, sensibility 70% and specificity 86%). Furthermore, by assessing clinical data and radiomic features in a combined model, the accuracy was not improved compared with the radiomic model alone.

Glioblastoma response to therapy is primary, especially in the early post-treatment setting; in that scenario Radiomics could be helpful in the detection of pseudoprogression and it was shown to yield promising results [63,64]. Kim J.Y. et al. [63] tested a quantitative approach to differentiate early cancer progression from pseudoprogression, by performing multiparametric MRI radiomic-model based on twelve radiomic features, extracted from 61 multiparametric MRI within twelve weeks after treatment. This model showed high performance in distinguishing pseudoprogression from early tumor progression with AUC of 0.90 (sensitivity 91.4% and specificity 76.9%). Similarly, Bani-Sadr F. et al. built two radiomic model, with and without clinical data including MGMT status, the latter model reaching the highest diagnostic accuracy (83%) [64].

## 9. Hematologic Disorders

Lymphoproliferative disorders (LPD) include a heterogeneous group of diseases characterized by a pathological proliferation of lymphocytes, circulating in the blood (leukemias), involving the bone marrow, infiltrating lymphoid or solid organ (lymphomas) [65]. Recent studies looked at the use of Radiomics in the diagnosis, prognosis and response to therapy in patients affected by LPD.

Regarding diagnosis, Radiomics was investigated in differentiating lymphoma from other primary malignancies achieving good accuracy making it a potential non-invasive assessment tool [66,67]. Kong Z. et al. [66] tested Radiomics in distinguishing primary central nervous system lymphoma from glioblastoma multiforme (GBM). Seventy-seven patients (24 lymphoma and 53 GBM) were retrospectively enrolled and radiomic features were extracted from ^18^F-FDG PET/CT; thirteen features resulted in statistical significance in differentiating the two groups with an AUC ranging between 0.971 and 0.998. Furthermore, Ma Z. et al. [67] revealed the high performance of CT-based Radiomics to differentiate Borrmann type IV gastric cancer (GC) from primary gastric lymphoma in a cohort of seventy patients (30 GC and 40 lymphoma). The authors obtained 183 statistically significant parameters (*p* < 0.05) for distinguishing the two different histotypes and the two main features (root_mean_square; sum_variance) were used to build a Rad-score.

Radiomics was also investigated as a tool of outcome prognostication, useful for the physicians in addition to traditional scores (i.e., International Prognostic Index, IPI), demonstrating good performance to predict patient outcome especially when Radiomics was applied to ^18^F-FDG PET [68,69,70]. Recently, Aide N. et al. [68] proposed a radiomic approach to predict 2-year event free survival in newly diagnosed diffuse large B cell lymphomas treated with immunotherapy. The authors analyzed baseline ^18^F-FDG PET of 132 patients, both in training and validation datasets, and among all features extracted only Long-Zone High-Grey Level Emphasis resulted a good predictor (AUC 0.69; Log-Rank *p* < 0.0001). Similarly, Mayerhoefer M.E. et al. proposed a radiomic prognostic models in patients affected by mantle cell-lymphoma based on ^18^F-FDG PET/CT, also by adding clinical data, to predict 2-year progression-free survival [69]. A total of 107 treatment-naïve patients were enrolled and Entropy and SUV_mean_ were significantly predictive of 2-year progression free survival on multivariable analysis (OR 1.131, *p* = 0.027 and OR 1.272, *p* = 0.013, respectively). Lower Entropy and SUV_mean_ seemed to be associated with longer PFS. However, they obtained the best PFS prognostication by integrating radiomic data, clinical and lab-tests, expressed according to Mantle cell lymphoma modified-IPI [70], in a single model (*p* = 0.005).

Regarding response to therapy, Radiomics has not been able to predict response to therapy yet [71]. Parvez A. et al. applied a radiomic segmentation on ^18^F -FDG PET/CT in 82 patients affected by aggressive non-Hodgkin lymphomas [71]. They obtained no significant results among high dimensional data in prediction of response to therapy (*p* > 0.05), while grey-level nonuniformity and kurtosis (*p* = 0.013 and *p* = 0.035, respectively) correlate with DFS and OS.

## 10. Bone

Traditional imaging plays a pivotal role in bone tumor detection, and is routinely used in patient management [72]. To date, only a few studies tested the role of Radiomics in the detection, differential diagnosis and response to therapy of bone neoplasms.

Among tumor detection and differential diagnosis, Radiomics was tested and reached good accuracy in differentiating benign from malignant tumors [73,74]. Yin P. et al. [73], who developed and validated a machine-learning CT-based method to differentiate sacral chondroma from giant cell tumor on 95 patients, divided in training and validation set, by performing manually volumetric segmentation. They selected three different methods for features selection (Relief, LASSO and Random Forest) and classification (generalized linear models (GLM), SVM and Random Forest), the best results were obtained by combining LASSO and GLM (AUC 0.984 on enhanced CT). In addition, enhanced CT outperformed unenhanced CT (*p* < 0.05). Xu R. et al. [74] applied texture analysis on ^18^F-FDG PET/CT to distinguish malignant from benign bone and soft-tissue tumors. The authors found that combining textural features, extracted from CT and PET, was superior to traditional SUV_mean_ (*p* = 0.0008, sensitivity 86.44%, specificity 77.27% and accuracy 82.52%).

Radiomics was also tested to assess the chemotherapy response and patient prognosis, by assessing delta-Radiomics and developing a radiomic nomogram, and it was showed to be a promising tool in evaluating prognosis and response to therapy [75,76]. Lin P. et al. [75] proposed a CT-based delta-Radiomics nomogram to evaluate pathological response in patients affected by high-grade osteosarcoma. Five-hundred and forty features were extracted from pre- and post-treatment CT of 191 patients, through LASSO logistic regression 8 features resulted to be significantly different between good-responder and non-responder (*p* < 0.001), with an AUC > 0.823 both in training and validation dataset. Similarly, Wu Y. et al. [76] developed a radiomic nomogram, by including clinical and radiomic features, with the aim to predict the survival of patients affected by osteosarcoma. They stated that Radiomics could be and adding tool for the physicians in these patients’ management.

Metastases from different primary malignancies still represent the majority of bone cancers and Radiomics was evaluated to identify primary PCa as well as risk-assessment of bone metastases at time of diagnosis [77,78]. In fact, Lang N. et al. [77] compared Radiomics with traditional ROI-method in investigating primary tumor and spine metastases on DCE-MRI. The radiomic approach was superior to the ROI-method (accuracy 0.81 and 0.71, respectively) and could guide clinicians in the workup of metastatic patients. Zhang W. et al. [78] looked at radiomic nomogram, combining clinical and radiomic data from mpMRI, to predict the incidence of bone metastases in newly diagnosed PCa. Radiomic nomogram performed well with an AUC > 0.92 both in the training and validation datasets.

## 11. Soft Tissue Tumors

Radiomics has been recently investigated as a non-invasive imaging biomarker in soft tissue sarcomas (STS), the main soft tissue malignancy, with the objective of assessing cancer grading (low-grade G1 and high-grade G2/G3), response to therapy and prognosis.

Regarding cancer grading, Radiomics was investigated as an imaging biomarker to differentiate low- from high-grade cancers and to predict tumor grading preoperatively. This approach resulted encouraging results [79,80]. Peeken J.C. et al. [79] developed an MRI-radiomic model to stratify STS into low- (G1) and high-grade (G2/G3) in a preoperative clinical setting. The authors performed a manual volumetric segmentation of the primary tumor both in the training (122 patients) and validation (103 patients) datasets. They obtained three promising radiomic models, based on fat saturation T2w and T1w after contrast medium injection sequences, able to differentiate low- from high-grade STS (AUC 0.69). Similarly, Zhang Y. et al. [80] tested Radiomics’ ability to predict tumor grading in 35 patients with histological diagnosis of STS. The superior result was obtained with support vector machine classification, which showed an AUC of 0.92 and accuracy of 0.88. A quantitative approach was also proposed to differentiate lipoma from liposarcoma by Vos M. et al. [81], who developed a radiomic model able to distinguish benign from malignant lesions (AUC 0.83, sensitivity 0.68 and specificity 0.84), one of the main challenges in soft tissue tumors imaging.

In relation to response to therapy and prognosis prediction, Radiomics was investigated by Crombé A. [82,83] et al., who tested this approach in predicting STS response to NAC, in risk assessment of metastatic relapse and in prediction of prognosis, yielding the best results when combining radiomic with clinical data. One study by Crombé A. et al. [82] tested Radiomics on 65 MRI (T1w, pre- and post-contrast medium and T2w). The best model was obtained with random forest classification on training dataset (50 patients) to predict early tumor response with an AUC of 0.86, accuracy 88.1%, sensitivity 94.1% and specificity 66.3%. Furthermore, another study by Crombé A. et al. [83] evaluated Radiomics as a prognostic tool in order to overcome the misclassification of myxoid round-cell, a relevant prognostic factor in liposarcoma, with traditional biopsy. They aimed to predict the presence of myxoid round-cell by quantifying lesion heterogeneity with a radiomic manual segmentation, performed on 35 patients MRI (T2w). The best performance was achieved by combining radiomic features with visual assessment with an AUC of 0.925 and concordance index of 0.937. Radiomics seems to be a promising tool in STS management, in particular in case of uncertain diagnosis as well as in the assessment of prognosis, however further investigations remain necessary.

## 12. Limitations

To date, lack of standardization, prospective studies and histologic validation represent several aspects which limit the use of Radiomics in clinical routine. Lack of standardization, both in terms of image acquisition and segmentation, is one of the main Radiomics challenges. Interscanner variability (e.g., different vendors and acquisition parameters) and segmentation methods (e.g., manual, semi-automatic and completely automatic) denote the major issues that could affect the standardization and reproducibility of analysis. In addition, radiomic results have to be validated by significant prospective multicentric studies, including large number of patients, able to mimic the application of Radiomics in routine clinical setting, then verifying the effective contribution to manage oncologic patients. Furthermore, histologic validation seems to be lacking and should be considered as a parameter of results reliability, by considering that one of the main challenges of Radiomics is to support and, in selected case, to replace histological analysis.

## 13. Future Perspectives

Radiomics might be seen as a future prospective tool in oncologic imaging, able to overcome subjective imaging evaluation by providing a quantitative parameter reflecting microenvironmental tumor architecture. Furthermore, radiomic parameters could help clinicians in early diagnosis, prognosis prediction before starting any therapy and to evaluate response to therapy. Radiomics could be seen as a helpful tool to tailor therapy per patient, with the aim to reinforce the role of imaging in the emerging field of personalized medicine. Then, Radiomics could support cancer patient workup by adding an objective and stable evaluation of medical images.

## 14. Conclusions

In conclusion, Radiomics should be seen as an imaging tool for oncologists in the new era of targeted patient-centered therapy by covering the gap between histological results and real microenvironmental heterogeneity. In addition, Radiomics could overcome the limit of subjective imaging based evaluation and provide an quantitative objective estimation of cancer heterogeneity and patient survival. However, the best results still were obtained by combining clinical data with radiomic parameters, supporting the idea that this recent imaging technique should be seen as a complementary tool for clinicians to have a full overview of cancer patients. Nevertheless, Radiomics is not the definitive solution to solve all management problems of oncologic patients, but it could help physicians in designing a workflow tailored specifically to each patient according to quantitative parameters and outcomes data. Furthermore, Radiomics needs to be validated and standardized in order to cover the main gaps currently present in the literature regarding image acquisition, image segmentation, feature extraction, feature selection and modeling, which represent major steps able to affect feature stability and analysis reliability. Future multicentric studies are necessary for testing and validating each radiomic phase, by sampling the widest possible range of variables, to mimic real clinical situation.

In the future, Radiomics should be seen as a consistent and objective tool for clinical trial settings and tailored therapy by performing an accurate assessment of patient prognosis at the moment of diagnosis and during course of treatment. Radiomics could in the future, become a routine application in the field of radiology however future studies are needed to solve questions regarding imaging acquisitions, segmentations, processing and post-processing data analysis.

## Figures and Tables

**Figure 1 cancers-13-02681-f001:**
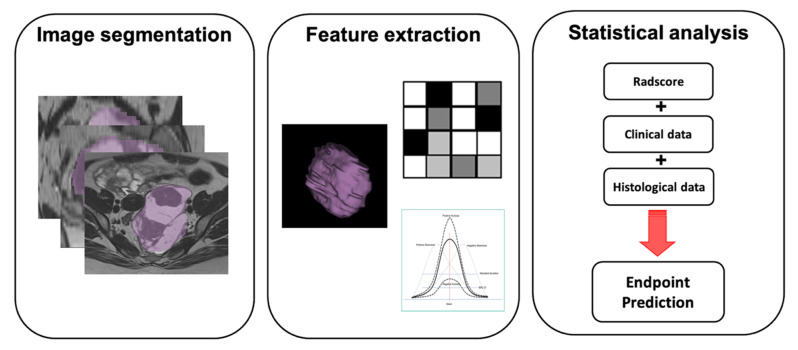
Ovarian cancer radiomic workflow. Image segmentation: ovarian cancer manually segmented on axial CT slice; feature extraction: tumor shape, texture and integration data; statistical analysis: combination of radiomic features with clinical and histological data to obtain predictive model.

**Figure 2 cancers-13-02681-f002:**
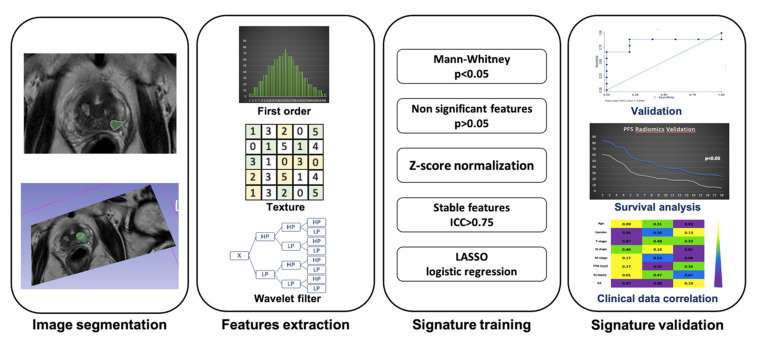
Radiomic analysis of prostate cancer. Image segmentation: prostate cancer segmented in a semi-automatic manner on axial T2, feature extraction: first order statistical features, texture features and wavelet filter, signature training: statistical analysis performed on mineable data extracted, signature validation: radiomic prognostic model validated on external cohort.

**Table 1 cancers-13-02681-t001:** Table summarizing the main studies of Radiomics applied to lung cancer.

Study	N Patients	Objective	Types of Evaluation	Model Performance	Imaging Modality	Features Selected	Nature of Study
Beig N. et al., Radiol. 2019 [7]	Total 290	AD vs. Granulomas	Intranodular and Perinodular radiomic analysis	0.80 0.76 0.60–0.61	CT	12	Multicentric Restrospective
Linning E. et al., Acad. Radiol. 2019 [8]	Total 278	SCLC vs. NSCLC SCLC vs. AD SCLC vs. SCC	Primary lesion radiomic analysis	End. 1 AUC: 0.74 End. 2 AUC: 0.82 End. 3 AUC: 0.66	CT	20	Monocentric Restrospective
Cong M. et al., Lung Cancer 2020 [9]	Training 455 Validation 194	Assessment nodal metastases	Predictive Radiomics on primary lesion	AUC: 0.91 AUC: 0.86	CT	7	Monocentric Restrospective
Zhang J. et al., Eur. J. Nucl. Med Mol. Imaging 2020 [10]	Training 175 Validation 73	EGFR status	Radiomic signature Fusion models	AUC: 0.86 AUC: 0.87	^18^F-FDG PET/CT	10	Monocentric Restrospective
Zerunian M. et al., Sci. Rep 2021 [11]	Total 21	OS PFS	Volumetric Textural analysis	End. 1 AUC: 0.72 End. 2 AUC: 0.74	CT	6	Monocentric Prospective
Khorrami M. et al., Lung Cancer 2019 [12]	Training 45 Validation 45	Pathological response	Intranodular and perinodular radiomic analysis	AUC: 0.90 AUC: 0.86	CT	13	Monocentric Restrospective

AD, Adenocarinoma; SCLC, small cell lung cancer; NSCLC, non-small lung cancer, SCC, squamous cell carcinomas; EGFR, epidermal growth factor receptor; OS, overall survival; PFS, progression free survival; SVM, support vector machine; ROC, receiver operating characteristic; CNN, convolutional neural networks; AUC, area under the curve; CT, computed tomography.

**Table 2 cancers-13-02681-t002:** Table summarizing the main studies regarding Radiomics applied to ovarian cancer.

Study	N Patients	Endpoint	Types of Evaluation	Model Performance	Imaging Modality	Features Selected	Nature of Study
Zhang H. et al., Eur. Rad. 2019 [26]	Validation 195 Testing 85	Benign vs. Malignant OEC type I vs. type II	LOO cross-validation Indipendent testing	End. 1 AUC: 0.97 End 1 AUC: 0.85 End 2 AUC: 0.96 End 2 AUC: 0.82	MRI	End. 1: 84 End. 2: 56	Monocentric Restrospective
Song X.L. et al., Eur. Rad. 2021 [27]	Training 72 Validation 32	Benign vs. Borderline Benign vs. Malignant Borderline vs. Malignant	2-class classification	End 1 AUC: 0.89 End 2 AUC: 0.86 End 3 AUC: 0.89	MRI	End. 1: 51 End. 2: 23 End. 3: 18	Monocentric Prospective
Meier A. et al., Abdom. Radiol. 2019 [28]	Total 88	Assosiation Survival and texture heterogeneity	Inter-site texture heterogeneity	*p* < 0.05	CT	3	Monocentric Restrospective
Lu H. et al., Nat. Commun 2019 [29]	Total 364	Survival	Radiomic prognostic vector	HR > 3.83	CT	4	Multicentric Restrospective
Himoto Y. et al., JCO Precis. Oncol. 2019 [30]	Total 75	Time to off-treatment	Intra-site texture heterogeneity Inter-site texture heterogeneity	*p* < 0.05 HR: 0.88 HR: 1.19	CT	7	Monocentric Restrospective
Danala. et al., Acad. Radiol 2017 [31]	Total 91	Early prediction treatment response	Delta Radiomics Fusion models	AUC: 0.77 AUC: 0.81–0.82	CT	24	Monocentric Restrospective

LOO, leave-one-out; AUC, area under the curve; HR, hazard ratio; MRI, magnetic resonance imaging; CT, computed tomography.

**Table 3 cancers-13-02681-t003:** Table summarizing the main studies regarding Radiomics applied to gliomas.

Study	N Patients	Endpoint	Types of Evaluation	Model Performance	Imaging Modality	Features Selected	Nature of Study
Tian Q. et al., J. Magn. Reson. Imaging 2018 [59]	Total 153	LGG vs. HGG Grade II vs. III	Volumetric radiomic analysis	End 1 AUC: 0.98 End 1 Acc: 96.8% End 2 AUC: 0.99 End 2 Acc: 98.1%	MRI	End. 1: 30 End. 2: 28	Monocentric Restrospective
Cho H.H. et al., PeerJ 2018 [60]	Total 285	LGG vs. HGG	Multi-regional radiomic features	AUC: 0.91 AUC: 0.88 AUC: 0.92	MRI	5	Multicentric Restrospective
Chang P. et al., Am. J. Neuroradiol. 2018 [61]	Total 259	IDH1 status 1p/19q codelation MGMT status	Volumetric deep learning CNN	End 1 Acc: 94% End 2 Acc: 92% End 3 Acc: 83%	MRI	64	Multicentric Restrospective
Li Z.C. et al., Eur. Radiol. 2018 [62]	Training 133 Validation 60	MGMT status	Multi-regional radiomic features Fusion models	AUC: 0.95 Acc: 87% AUC: 0.88 Acc: 80%	MRI	6	Multicentric Restrospective
Kim J.Y. et al., Neuro Oncol. 2019 [63]	Total 61	Pseudoprogression vs. Progression	Multiparametric radiomic models	AUC: 0.90	MRI	12	Monocentric Restrospective
Bani-Sadr. et al., Neurooncol. Adv. 2019 [64]	Training 55 Validation 21	Pseudoprogression vs. Progression	Multi-regional radiomic features	AUC: 0.82 Acc: 83% AUC: 0.85 Acc: 79.2%	MRI	11	Monocentric Restrospective

LGG, low-grade gliomas; HGG, high-grade gliomas, IDH1, isocitrate dehydrogenase 1; MGMT, O(6)-Methylguanine-DNA methyltransferase SVM, support vector machine; ROC, receiver operating characteristic; CNN, convolutional neural networks; AUC, area under the curve; Acc; accuracy; MRI, magnetic resonance imaging.

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
