# Peer review of "Radiomics in Oncology, Part 2: Thoracic, Genito-Urinary, Breast, Neurological, Hematologic and Musculoskeletal Applications"

_cancers, 2021, doi:10.3390/cancers13112681_

Round 1
Reviewer 1 Report
General comments:
In the current paper Caruso et al. provide a review on the applications of radiomics on thoracic, genito-urinary, breast, neurological, hematologic and musculoskeletal cancers. It seems that this is Part 2 of a 2 manuscript review. Just for context, I should clarify that I was not a referee for Part 1, which I could not find online either.
Focusing only on the current manuscript, the review focuses on the aforementioned types of cancers, discussing briefly some of the works published for each type of cancers. First thing I miss is that the paper does not have an introduction. The paper does not even defines radiomics, which is, in fact, a very wide field, so what we are discussing when we discuss radiomics in the current work should be stated.
Regarding the description of the applications on different cancers, I find the writing of the different sections confusing and difficult to read. I like the presentation divided in “diagnostics”, “prognosis”, “response to therapy” in some sections (i.e. Ovarian Cancer). This could be expanded to all the sections.
The English level is very poor, I would recommend that the manuscript is revised by a native English speaker with a scientific background before publication.
Please be consistent with the usage of terms like Radiomics, radiomics, radiomic, etc… In my opinion you should use the term “radiomics”, without capitals.
I feel that most of the presented data in form of AUCs, etc is uncontextualized. In the case of a qualitative review, I would center in explaining the problems radiomics is trying to solve, which are the approaches used, and how they compare with current techniques, etc... In this regard, the paper might cover a wider number of papers. The number of references dealt with seems small to me. The manuscript tends to focus on commenting individual papers, but this does not create a context on a general application. It would be preferable to reference a group of works dealing with the same problem, and try to extract some general conclusions.
I think that the manuscript needs some improvements before publication. For illustrating most of my general comments, please find some specific comments below:
Title:
Please, write Oncology, Thoracic, Genito-urinary, etc… without capital leters. Replace application with applications.
Simple Summary:
“Some the main” à “Some of the main”.
“application” à “applications”
Plenty of other grammatical and orthographical errors. Poor writing. Please revise.
Abstract:
The abstract seems to summarize the previous paper not this one. For example:
- The abstract seems to highlight heterogeneity, but this concept is not discussed in the current manuscript.
- The abstract highlights “the significant methodology limitations and lack of validation”, but these are not discussed in the current manuscript.
It seems contradictory to affirm that radiomics has a “pivotal role” in oncological translational imaging. Radiomics is still an experimental, incipient field. Please consider a correction.
“Recently, Radiomics has been extensively applied on lung cancer and literature accounts for hundreds of studies [1-3] (Figure 1), focus on diagnosis, prognosis and response to treatment.” The references is misplaced. Please provide references for the diagnosis, prognosis, and response to treatment. Ensure that the references are relevant. I don’t think Figure 1 is correctly linked here, and I am not sure if the Figure is original. If taken from elsewhere, please reference accordingly.
“Performance of Radiomics was compared with Convolutional Neural Network (CNN) and visual assessment”. CNN is just a way to construct a model. It is perfectly possible to generate a CNN model for predicting (i.e. treatment outcome) trained by using radiomics features. Please clarify.
“In future clinical settings, an additional Radiomics area application could be represented from stratification patient risk”. This sentence makes no sense.
The commented paper from Kan Y. seems to extract textural features from the patient lymph nodes rather than the primary tumor. This seems to be a different application from others on the review, i.e. link radiomics with heterogeneity and heterogeneity to tumor aggressiveness.
“Results underlined that radiomic signature allowed better accuracy to discriminate LN in comparison with radiologist qualitative assessment (AUC 0.753 and 0.754, accuracy 0.75 and 0.72 vs. 0.65 and 0.49, in test and validation cohorts, respectively)”. I am unsure that these listings of AUCs not compared to alternative methods are an addition of interest for readers.
Author Response
Dear Editor, Dear Reviewers,
We would like to sincerely thank you for the detailed review Part II or our manuscript, your valuable suggestions, and your precise comments.
Please find responses to your comments and consecutive changes based on your recommendations below.
Thank you again.
Reviewer’s comments are in bold. Changes in the manuscript are in italics.
Reviewer 1
In the current paper Caruso et al. provide a review on the applications of radiomics on thoracic, genito-urinary, breast, neurological, hematologic and musculoskeletal cancers. It seems that this is Part 2 of a 2 manuscript review. Just for context, I should clarify that I was not a referee for Part 1, which I could not find online either.
Reply: Sincerely thank you for the comment, the present manuscript is the second part; we have submitted the Part 1 entitled “Radiomics in Oncology, Part 1: Technical Principles and Gastrointestinal application” in the same special issue “Cancer Imaging, Current Practice and Future Perspective”. Unfortunately, Part 1 and Part 2 have been assigned to different reviewers and Part I is actually submitted as R1 after minor revision. We hope you will see it online soon, however, if you want, we will provide you full details regarding Part 1.
Focusing only on the current manuscript, the review focuses on the aforementioned types of cancers, discussing briefly some of the works published for each type of cancers. First thing I miss is that the paper does not have an introduction. The paper does not even defines radiomics, which is, in fact, a very wide field, so what we are discussing when we discuss radiomics in the current work should be stated.
Reply: We want to thank for your valuable suggestion, the Introduction has been added following your precise comment. Regarding the definition of Radiomics, Part 1 provide a full overview about technical principles or Radiomics; thus we prefer to leave the definition of Radiomics in Part 1.
Regarding the description of the applications on different cancers, I find the writing of the different sections confusing and difficult to read. I like the presentation divided in “diagnostics”, “prognosis”, “response to therapy” in some sections (i.e. Ovarian Cancer). This could be expanded to all the sections.
Reply: Sincerely thank you for the important suggestion, we have modified several sections by adding a structured division in the main radiomics application fields for each organ and/or system, according to your insight.
The English level is very poor, I would recommend that the manuscript is revised by a native English speaker with a scientific background before publication.
Reply: We want to thank for your valuable suggestion, the English has now been deeply revised by Dr Damiano Caruso, MD, PhD, who worked a research fellow for a year and half at Medical University of South Carolina, MUSC, Charleston, South Carolina, United States. After that experienced he has obtained C2 level (proficient) at EF SET, as you can confirm from the attached certificates. In addition, Prof Andrea Laghi has the scientific background requested (https://scholar.google.it/citations?user=ieyzLswAAAAJ&hl=en).
Please be consistent with the usage of terms like Radiomics, radiomics, radiomic, etc… In my opinion you should use the term “radiomics”, without capitals.
Reply: Thank you for your precise comment. Unfortunately, in the Part I of our paper we have used “Radiomics” as substantive and “radiomic” as adjective. Then, we have decided to maintain these forms also in this Part II and we hope you will accept to leave it as it is, otherwise we will be happy to modify Part I and II accordingly.
I feel that most of the presented data in form of AUCs, etc is uncontextualized. In the case of a qualitative review, I would center in explaining the problems radiomics is trying to solve, which are the approaches used, and how they compare with current techniques, etc... In this regard, the paper might cover a wider number of papers. The number of references dealt with seems small to me. The manuscript tends to focus on commenting individual papers, but this does not create a context on a general application. It would be preferable to reference a group of works dealing with the same problem, and try to extract some general conclusions.
Reply: We want to apologize for the presentation of uncontextualized data. We have improved the manuscript by describing the main issues of conventional imaging for each section, in order to create a context on a general Radiomics application for each group of paper presented with proper general conclusions. We hope that these are sufficient improvements.
I think that the manuscript needs some improvements before publication. For illustrating most of my general comments, please find some specific comments below:
Reply: Thank you for your precise and accurate suggestions.
Title:
Please, write Oncology, Thoracic, Genito-urinary, etc… without capital leters. Replace application with applications.
Reply: We want to thank you for the comment, we have modified the Title according to your accurate suggestions.
Simple Summary:
“Some the main” à “Some of the main”.
“application” à “applications”
Plenty of other grammatical and orthographical errors. Poor writing. Please revise.
Reply: We want to thank for your valuable corrections, this section has been improved.
Abstract:
The abstract seems to summarize the previous paper not this one. For example:
- The abstract seems to highlight heterogeneity, but this concept is not discussed in the current manuscript.
- The abstract highlights “the significant methodology limitations and lack of validation”, but these are not discussed in the current manuscript.
Reply: We want to thank for your helpful comments, the abstract section has been improved by avoiding topics not covered in Part II.
It seems contradictory to affirm that radiomics has a “pivotal role” in oncological translational imaging. Radiomics is still an experimental, incipient field. Please consider a correction.
Reply: We apologize for the confounding expression regarding role of radiomics, it has been corrected.
“Recently, Radiomics has been extensively applied on lung cancer and literature accounts for hundreds of studies [1-3] (Figure 1), focus on diagnosis, prognosis and response to treatment.” The references is misplaced. Please provide references for the diagnosis, prognosis, and response to treatment. Ensure that the references are relevant. I don’t think Figure 1 is correctly linked here, and I am not sure if the Figure is original. If taken from elsewhere, please reference accordingly.
Reply: We want to apologize for misplacing of the references, we have removed those and mentioned the references discussed in the section. Thanks for the comment concerning the Figure 1. All the Figures provided in the manuscript are original. In particular, Figure 1 is a graphical representation of image acquisition, segmentation and feature extraction of small lung cancer studied in our Hospital and we provided to segment the lesion and to assemble the figure to give schematic representation of the process. If needed we can provide the anonymized images of the case and the image draft of the Figure 1. However, we have removed this according to the suggestion of Reviewer 2 in order to avoid the redundancy with the other figures. Then, we have added the Table 1 summarizing the major studies focused on lung cancer. We hope that the changes are satisfactory.
“Performance of Radiomics was compared with Convolutional Neural Network (CNN) and visual assessment”. CNN is just a way to construct a model. It is perfectly possible to generate a CNN model for predicting (i.e. treatment outcome) trained by using radiomics features. Please clarify.
Reply: We want to thank for your valuable annotation, the confounding sentence has been modified accordingly.
“In future clinical settings, an additional Radiomics area application could be represented from stratification patient risk”. This sentence makes no sense.
Reply: Thank you for the punctual suggestion, we have modified the sentence accordingly.
The commented paper from Kan Y. seems to extract textural features from the patient lymph nodes rather than the primary tumor. This seems to be a different application from others on the review, i.e. link radiomics with heterogeneity and heterogeneity to tumor aggressiveness.
Reply: Thank you for your valuable comment, the paper mentioned is relative to nodal assessment in patients affected by cervical cancer. Nodal metastases could substantially affect patient staging, prognosis, and therapeutic approach. Conventional imaging has several limits in identify metastatic nodes before surgery, then radiomics could represent a consistent promising tool to face this issue and manage these patients properly.
“Results underlined that radiomic signature allowed better accuracy to discriminate LN in comparison with radiologist qualitative assessment (AUC 0.753 and 0.754, accuracy 0.75 and 0.72 vs. 0.65 and 0.49, in test and validation cohorts, respectively)”. I am unsure that these listings of AUCs not compared to alternative methods are an addition of interest for readers.
Reply: We would like to thank for the observation, we rewrote the results of the paper with a proper discussion by avoiding the only listing of data.
I wish you all the best,
Sincerely,
Prof Andrea Laghi and co-autho

Reviewer 2 Report
The manuscript summarizes Radiomics studies in different oncologic applications, including thoracic, genitourinary, breast, neuro, hematologic and musculoskeletal applications. Although the paper comprehensively included many different Radiomics studies, the authors should provide analysis and interpretation of these studies by explaining their relevance and significance.
Technical descriptions for each application (Figure 1-3) were somewhat duplicative and redundant. It will be easier if the paper describes the overall Radiomics analysis framework, including image acquisition, pre-processing, feature extraction, feature selection, and statistical modeling.
Tables summarizing the major results would be helpful to understand their relevance and differences. The table could include 1) a number of cases used for training and testing, 2) types of evaluation (e.g., cross-validation or independent testing), 3) overall model performance, 4) imaging modality, 5) a number of selected radiomics features, and 6) single- or multi-center study.
Current challenges and future studies for each application should be included in the Discussion, such as the generalizability of the radiomics model and issues related to pre-processing (image normalization, registration, and segmentation) and feature selection.
Author Response
Dear Editor, Dear Reviewers,
We would like to sincerely thank you for the detailed review Part II or our manuscript, your valuable suggestions, and your precise comments.
Please find responses to your comments and consecutive changes based on your recommendations below.
Thank you again.
Reviewer’s comments are in bold. Changes in the manuscript are in italics.
Reviewer 2
The manuscript summarizes Radiomics studies in different oncologic applications, including thoracic, genitourinary, breast, neuro, hematologic and musculoskeletal applications. Although the paper comprehensively included many different Radiomics studies, the authors should provide analysis and interpretation of these studies by explaining their relevance and significance.
Reply: Thank you so much for your overall evaluation of the manuscript and the suggestions. We have modified the sections accordingly by providing some interpretation for each group of paper presented to avoid that these seems to be uncontextualized. We hope the quality of manuscript has been satisfactory improved.
Technical descriptions for each application (Figure 1-3) were somewhat duplicative and redundant. It will be easier if the paper describes the overall Radiomics analysis framework, including image acquisition, pre-processing, feature extraction, feature selection, and statistical modeling.
Reply: We want to thank for the annotation. We have modified the figures by eliminating the Figure 1 that seems too reductive. However, in the Part I we dedicated a section in which the Radiomics analysis framework has been explained. You can find below the section:
2. Technical Principles
Radiomics workflow is divided into multi-steps, images acquisition, image segmentation, features extraction, features selection, model construction and validation. Each step is dependent on the previous one and the aim is to obtain a performant and reliable prognostic model, usually driven by Artificial Intelligence, a research field in which the human intelligence is mimicked by mathematical and statistical approaches able to create performant artificial neural networks [10]. Machine learning, neural network and deep learning are some Artificial Intelligence subfields used for applying mathematical and statistical approaches to Big Data (e.g. radiomic features, clinical data, disease free survival, survival) with the ability to find and interpretate occult models, built on mineable data and their interpretation needed for clinical support [11]. To standardize Radiomics approach in routine clinical setting a structured and robust workflow is requested for achieving reliable and consistent data.
The first step is based on images acquisition, one of the most challenging aspects due to the lack of protocol and parameters standardization and to the facts that it affects the reproducibility of analysis, particularly important in multicenter center studies [12]. Several options have been recently proposed to overcome the bias of acquisition protocols by performing a test-retest analysis, with the goal of eliminating radiomic features affected by higher variability and maintaining only robust features [13]. To face this issue an automatic acquisition protocol has been proposed among several CT scanners and these methods showed some promising results in terms of reaching robustness in radiomic parameters by using resampling approach to uniform each voxel size as post-acquisition correction. However, this approach has not been able to modulate acquisition specifications for each different CT scanner at the same time and it could be difficult to use in retrospective studies routinely [14]. Regarding test-retest analysis, some relevant results have been obtained by analyzing the reproducibility of radiomic parameters among different CT scanners, both by changing CT specifications (intra-analysis) and by comparing different CT scanners (inter-analysis), and it was showed that high number of radiomic parameters could be altered by changing of some CT acquisition parameters [15]. However, these results were performed in phantom studies, in which the inter-patient variability has not been assessed, thus future investigations are needed to identify the best approach to standardize the image acquisition. Nevertheless, the robust features often resulted to have low diagnostic performance [16]. In addition, the applying of a post-reconstruction batch harmonization has been also proposed to reduce the variability among centers by using global scaling, in which signal intensities are harmonized by eliminating the mean and the unit variance is downsized, z-standardization, where each feature is normalized considering the mean and standard deviation by providing some comparable results, and histogram-matching, by transforming intensity histogram in order to combine them and find the reference histogram [17]. However, each method has different advantages and drawbacks, and they should be weighted in different clinical scenarios.
The second phase is image segmentation, in which regions of interest (ROIs) are outlined around cancer tissues, by covering the entire lesion area and avoiding some unnecessary structures (i.e. vessels, biliary duct, healthy parenchyma) that could alter the heterogeneity analysis. The image segmentation is the leading cause of features variability, with a specific focus on inter-reader lack of reproducibility due to high variability of extracted features without a specific feature selection. High variability could affect the consistence of radiomic signature; thus the reproducible features should be selected, excluding features classified as consistently unstable. To evaluate the robustness of radiomic features might be used the interclass correlation coefficient (ICC) among several datasets and it has been showed that a radiomic feature with high ICC on one dataset it will be robust and stable also in the others [18,19]. Radiomics parameters are extracted from image volumes by performing manually, semi-automatic, or completely automatic segmentation. The first method is a time-consuming process, usually affected by high intra and inter-observer variability also according to radiologist experience, then difficult to apply in clinical setting as routine [20]. Semi-automatic and automatic segmentation process are demonstrated to be promising in homogeneous lesions, where interaction of external reader was minimum, with high accuracy, low inter-reader variability, and high reproducibility [21]. Since radiomic analysis could be highly affected by these different methods of segmentation, then the validation, standardization and robustness are needed [22-24]. Consequently, the assessment of Radiomics features variability related to segmentation processes has been emerging with the proposal to eliminate the features with high variability and low prognostic strength.
Feature extraction represents the third step of Radiomics process. Quantitative features have to be extracted from ROIs previously outlined on tissues of interest (Figure 2). Features obtained are divided into shape features, describing the shape and geometry of ROIs (i.e. volume, maximum surface), first order statistics features, resulting from grey-level histogram and describing voxel values without consider the relationship with other voxels, second order statistics features (i.e. gray-level co-occurrence matrices or gray-level run length matrices), derived by analyzing each pixel and its relationship with those adjacent in specific matrices, and high order statistics features, resulted from mathematical algorithm after the application of specific filters (Table 1).To avoid some unnecessary bias, affecting data homogeneity, it is critical to standardize each step of analysis [25].
Feature selection is one of the main key steps of Radiomics process. It is related to the selection of best performing parameters among the large quantity of parameters extracted, often interconnected and characterized by overfitting. In that context, a specific and punctual selection of features obtained seems to be necessary to avoid some bias in model construction; it is essential to define the endpoint of analysis with precise clinical applications to select the best performing features with some dedicated approaches. Two most unsupervised approaches used to perform features selection are cluster analysis and principal component analysis. The first is based on clustering similar radiomic parameters according to high cluster redundancy and low inter-cluster correlation, usually illustrating as cluster heat map; and only one parameter from each cluster is selected for further analysis. The principal component analysis is built on creating small set of non-correlated features extracted from a large amount of correlated features with the aim to explain the total variable variation by using the smallest number of features, usually illustrating as score-plots [1]. The goal is to select the best features assessed as non-redundant, reproducible and performing.
The next step is the model building, in which the best selected features, clinical data, and histological data are combined, to assess the pre-fixed outcome (e.g. survival, disease free progression, therapeutic assessment). In this critical step several approaches have been proposed to perform multivariate analysis, tailored per endpoint, that could provide a clinical tool able to support clinical decision-making. Different statistical methods and data mining and/or machine-learning methods were investigated. The main classifiers used in practice are random forest, linear regression, logistic regression, Cox proportional hazards regression, least absolute shrinkage and selection operator (LASSO), support vector machines (SVM), neural network, deep learning and decision tree [26]. To date, it was shown that there is no unique best classifier, but each of them should be evaluated in order to obtain the consistent, reliable and generalizable models [27].
In conclusion, the constructed model must be trained and optimized on training and testing set, then validated on external cohort in order to obtain a reliable model on different patient groups. Athought the validation on external datasets is necessary, sometimes it could be not possible. Then, several strategies have been tested to overcome this limit by obtaining an internal validation (i.e. random subsampling and nested approach), but the main issue of these approaches is the risk to alter the features selection algorithm and the results obtained could be unreasonably optimistic [28]. To sum up, the validation model achieved through external validation is the goal standard and the comparability of radiomic features extracted from medical images, acquired with different protocols, and segmentation processes performed with different tasks are the main challenges of Radiomics approaches.”
Tables summarizing the major results would be helpful to understand their relevance and differences. The table could include 1) a number of cases used for training and testing, 2) types of evaluation (e.g., cross-validation or independent testing), 3) overall model performance, 4) imaging modality, 5) a number of selected radiomics features, and 6) single- or multi-center study.
Reply: We want to thank for the valuable and accurate annotation. We have added three Tables summarizing the major studies focused on lung cancer, ovarian cancer, and neurological system. We hope that these could help the reader and could be satisfactory.
Current challenges and future studies for each application should be included in the Discussion, such as the generalizability of the radiomics model and issues related to pre-processing (image normalization, registration, and segmentation) and feature selection.
Reply: Thank you for your precise and helpful comment. We have expanded the Conclusion in order to discuss the major radiomic challenges and the need of standardization to apply this approach as clinical routine.
I wish you all the best,
Sincerely,
Prof Andrea Laghi and co-authors.

Reviewer 3 Report
Dear Authors,
it is a nice and complete review about radiomics in some oncology applications.
The paper is well structured and written. Only line 330 need to be reformulate for my taste.
All cited papers are valuable and up to date.
Probably it is a part of another paper (since I not read part I nor part III) but, if it is not the case, it would be nice to have a discussion about the limitations of radiomic approaches (number of cases, single vs. multicentre study, single vs. multiple imaging protocols, single vs. multivendor imaging modalities etc.).
Best Regards
Author Response
Dear Editor, Dear Reviewers,
We would like to sincerely thank you for the detailed review Part II or our manuscript, your valuable suggestions, and your precise comments.
Please find responses to your comments and consecutive changes based on your recommendations below.
Thank you again.
Reviewer’s comments are in bold. Changes in the manuscript are in italics.
Reviewer 3
Dear Authors,
it is a nice and complete review about radiomics in some oncology applications.
The paper is well structured and written. Only line 330 need to be reformulate for my taste.
All cited papers are valuable and up to date.
Reply: Thank you so much for your overall evaluation of the manuscript and the suggestion. The paper has been improved accordingly.
Probably it is a part of another paper (since I not read part I nor part III) but, if it is not the case, it would be nice to have a discussion about the limitations of radiomic approaches (number of cases, single vs. multicentre study, single vs. multiple imaging protocols, single vs. multivendor imaging modalities etc.).
Reply: We want to thank for the comment. As you correctly supposed, the present manuscript is the second part; we have submitted the Part 1 entitled “Radiomics in Oncology, Part 1: Technical Principles and Gastrointestinal application” in the same special issue “Cancer Imaging, Current Practice and Future Perspective”. Unfortunately, Part 1 and Part 2 have been assigned to different reviewers and Part I is actually submitted as R1 after minor revision. We hope you will see it online soon. Part 1 has a dedicated section with main limitations and proposals to overcome them. However, if you want, we will provide you full details regarding Part 1.
I wish you all the best,
Sincerely,
Prof Andrea Laghi and co-authors.

Round 2
Reviewer 1 Report
In the current paper Caruso et al. provide a review on the applications of radiomics on thoracic, genito-urinary, breast, neurological, hematologic and musculoskeletal cancers. This is a second version of the manuscript after a first round of reviews. The authors have answered to most of the provided comments. In addition, I may also highlight that some interesting additions derived from other referee’s comments that increase the overall quality of the manuscript. In general, I can see that the manuscript does a better work as a qualitative review now, and that the readability and organization of the sections has improved.
Nevertheless, I think that the manuscript still requires some improvements before publication. I provide some general and specific comments.
General comments:
- In the previous review I have pointed that the manuscript shall be reviewed by a native English speaker. The authors have improved the manuscript overall. An answer is provided pointing that one of the authors has worked for a year and half in the US and has obtained a C2 qualification, which I do not doubt. My level of English is not better, and I am not a native speaker either. Nevertheless, I was able to notice that there are still plenty of grammar and expression errors and inaccuracies, which must be fixed. For a glimpse, please see in the specific comments potential corrections to the first 75 lines. I shall highlight that this is a review, so readability must be excellent. If the reviewers can not appeal to a native speaker colleague, consultancy services that provide such services exist and are cost affordable, I have used them on several occasions.
- Regarding the usage of the terms Radiomics, radiomics, radiomic, the authors have exposed that they have used “Radiomics” as substantive and “radiomic” as adjective. I accept this, there is no need to modify Part I. Still, I found some inconsistencies in the usage. See for example line 68, “Radiomics features” should be “radiomic features” following this convention, or line 86, “radiomic and 86 combined radiomic models” should be “Radiomics and combined radiomics models”, and many others.
- Regarding the comment about the paper from Kan Y, I am aware of the relevance of the nodal status for staging and prognosis, and of the limitations of conventional imaging. Due to this, precisely, I was asking you to expand on this topic. As you might agree, extracting radiomic features is different to conventional measurements from the primary tumor, even those related with nodal status prediction, such as De Bernardi et al. ( https://pubmed.ncbi.nlm.nih.gov/30136163/), and deserve a separate comment. I would like to see a sentence or paragraph such as “In addition to radiomic features extracted from the primary tumor, other authors have focused in evaluating measurements from lymph nodes [Ref]. ….” This applies also to other sections where similar approaches are discussed.
- I feel that the included figures are redundant. See specific comments. In Tables 1 and 3 the authors provide in the column “type of evaluation” terms such as “SVM”, Logistic regression” or “random forest”. As I stated in the previous review, these are NOT radiomics approaches. This are just types of modeling approaches in machine learning, that might be applied with or without extracting radiomic features (i.e., is perfectly possible to train a support vector machine with clinical data only). In my opinion, the different classification used in Table 2 is more adequate.
Specific comments:
Avoid the formulation “John Doe and colleagues” when you cite a work. Please attach to “Doe et al. []”
Title: Ok
Simple summary:
Line 12: Imaging -> imaging
Line 12-13: Prediction prognosis -> “prognosis prediction”
Line 14: “In this Part II have been described” -> “In this part II we described”
Abstract:
Line 18: Prediction prognosis -> “prognosis prediction” or “predicting prognosis” if reprhasing is done
Line 19: “several studies enhanced Radiomics as a skill in oncologic imaging” -> I don’t think that enhanced is the correct verb for the sentence. In addition, I think that Radiomics is not a “skill”
Introduction:
Line 30: “Radiomics has been emerging in the workup of cancer, this imaging landscape 30 was extensively investigated in tumor diagnosis”. This sentence makes no sense.
“The lack of standardization limits the routine use of Radiomics as a clinical tool, although the increasing number of significative results support the future radiomic application in 39 cancer patient management”. Please add some references.
Lung Cancer:
Line 48: focus on -> “focusing on”
Line 49: (Table 1). Please add text describing the table. i.e. “Table 1 summarizes….”
Line 57. Review this sentence. i.e. Regarding lung cancer diagnosis, several works have investigated the performance of Radiomics in distinguishing …. (add references, Beit et al. are not the only ones investigating this.)
Line 59: Consider using also SCLC abbreviation.
Line 66: against a Convolutional… and visual assessment by
Line 67: showed a higher performance of the combined… This “missing articles” issue is recurrent across the whole text.
Line 67-68: Following the stablished convention should be “radiomic features”
Line 70: A similar approach was proposed
Lines 74-75: Resulting in a better … . In the same line, between over and CT please remove the double spacing.
From this point on I will not highlight all grammar or English errors, just highlighted some to show the authors that English improvements are still required.
Line 140: “in the French and Canadian cohorts”. First time you mention that there are French and Canadian cohorts…
Lines 167-169: You should comment the Table and Figure in the main text. In the Figure caption, please correct “lung cancer”.
Lines 181-184: The list of features provides no value unless it can be contextualized (i.e. compared with the results of other works) or they where validated in other works (i.e. using a different database).
Line 204: “Moving to prognosis, several studies has been performed up to now and some sig nificant correlation has been found between radiomic and outcome data, but not relevant 205 results has been demonstrated in identifying BRCA mutation panel”. Please rephrase.
Line 222: in which Radiomics showing -> No verb… “in which Radiomics showed”?.
Line 244: Figure 2. I do not understand what this Figure contributes to. It seems to be the same Figure than 2 just replacing Ovarian with Prostate cancer.
Line 247: an helpful method. -> a helpful method.
Line 279: Again, “Radiomics signature". According to the established convention, this should be “radiomic signature”
Line 298: “in assessment" -> in the assessment. In addition, shall rephrase as “tumor grading and local invasion, and only few studies tested Radiomics”, I do not think that the next item is part of the same list.
Author Response
Dear Editor, Dear Reviewers,
We would like to sincerely thank you for the detailed review Part II or our manuscript, your valuable suggestions, and your precise comments.
Please find responses to your comments and consecutive changes based on your recommendations below.
Thank you again.
In the current paper Caruso et al. provide a review on the applications of radiomics on thoracic, genito-urinary, breast, neurological, hematologic and musculoskeletal cancers. This is a second version of the manuscript after a first round of reviews. The authors have answered to most of the provided comments. In addition, I may also highlight that some interesting additions derived from other referee’s comments that increase the overall quality of the manuscript. In general, I can see that the manuscript does a better work as a qualitative review now, and that the readability and organization of the sections has improved.
Nevertheless, I think that the manuscript still requires some improvements before publication. I provide some general and specific comments.
Reply: We want to thank you for the appreciable evaluation.
General comments:
- In the previous review I have pointed that the manuscript shall be reviewed by a native English speaker. The authors have improved the manuscript overall. An answer is provided pointing that one of the authors has worked for a year and half in the US and has obtained a C2 qualification, which I do not doubt. My level of English is not better, and I am not a native speaker either. Nevertheless, I was able to notice that there are still plenty of grammar and expression errors and inaccuracies, which must be fixed. For a glimpse, please see in the specific comments potential corrections to the first 75 lines. I shall highlight that this is a review, so readability must be excellent. If the reviewers can not appeal to a native speaker colleague, consultancy services that provide such services exist and are cost affordable, I have used them on several occasions.
Reply: Thank you for your supporting annotation. The text has been revised and edited by Dr. Marwen Eid, a native American English, working at Northwell Health Staten Island University Hospital, Staten Island, New York, USA, 10305. In addition, we have added him as an Author of the paper. We hope that the linguistic revision will be now satisfactory.
- Regarding the usage of the terms Radiomics, radiomics, radiomic, the authors have exposed that they have used “Radiomics” as substantive and “radiomic” as adjective. I accept this, there is no need to modify Part I. Still, I found some inconsistencies in the usage. See for example line 68, “Radiomics features” should be “radiomic features” following this convention, or line 86, “radiomic and 86 combined radiomic models” should be “Radiomics and combined radiomics models”, and many others.
Reply: Thanks for your precise and valuable annotation, we have modified and revised the text accordingly.
- Regarding the comment about the paper from Kan Y, I am aware of the relevance of the nodal status for staging and prognosis, and of the limitations of conventional imaging. Due to this, precisely, I was asking you to expand on this topic. As you might agree, extracting radiomic features is different to conventional measurements from the primary tumor, even those related with nodal status prediction, such as De Bernardi et al. ( https://pubmed.ncbi.nlm.nih.gov/30136163/), and deserve a separate comment. I would like to see a sentence or paragraph such as “In addition to radiomic features extracted from the primary tumor, other authors have focused in evaluating measurements from lymph nodes [Ref]. ….” This applies also to other sections where similar approaches are discussed.
Reply: We want to thank for your precise and accurate comment, we have modified the sections accordingly and added the suggested reference (PMID 30136163).
- I feel that the included figures are redundant. See specific comments. In Tables 1 and 3 the authors provide in the column “type of evaluation” terms such as “SVM”, Logistic regression” or “random forest”. As I stated in the previous review, these are NOT radiomics approaches. This are just types of modeling approaches in machine learning, that might be applied with or without extracting radiomic features (i.e., is perfectly possible to train a support vector machine with clinical data only). In my opinion, the different classification used in Table 2 is more adequate.
Reply: Thank you for your appropriate annotation, we have improved and modified the Tables 1 and 3 according to your suggestions improving the readability.
Specific comments:
Avoid the formulation “John Doe and colleagues” when you cite a work. Please attach to “Doe et al. []”
Reply: We want to thank for your valuable comment, we have modified the paper accordingly.
Title: Ok
Simple summary:
Line 12: Imaging -> imaging
Line 12-13: Prediction prognosis -> “prognosis prediction”
Line 14: “In this Part II have been described” -> “In this part II we described”
Reply: Thank you for your precise and valuable comments, we have modified the Simple summary section accordingly.
Abstract:
Line 18: Prediction prognosis -> “prognosis prediction” or “predicting prognosis” if reprhasing is done
Line 19: “several studies enhanced Radiomics as a skill in oncologic imaging” -> I don’t think that enhanced is the correct verb for the sentence. In addition, I think that Radiomics is not a “skill”
Reply: Thank you so much for your annotations. The Abstract has been improved accordingly.
Introduction:
Line 30: “Radiomics has been emerging in the workup of cancer, this imaging landscape 30 was extensively investigated in tumor diagnosis”. This sentence makes no sense.
“The lack of standardization limits the routine use of Radiomics as a clinical tool, although the increasing number of significative results support the future radiomic application in 39 cancer patient management”. Please add some references.
Reply: Thank you for your valuable suggestions, the Introduction section has been improved and some references have been added (PMID 32785796 and 32575082).
Lung Cancer:
Line 48: focus on -> “focusing on”
Line 49: (Table 1). Please add text describing the table. i.e. “Table 1 summarizes….”
Line 57. Review this sentence. i.e. Regarding lung cancer diagnosis, several works have investigated the performance of Radiomics in distinguishing …. (add references, Beit et al. are not the only ones investigating this.)
Line 59: Consider using also SCLC abbreviation.
Line 66: against a Convolutional… and visual assessment by
Line 67: showed a higher performance of the combined… This “missing articles” issue is recurrent across the whole text.
Line 67-68: Following the stablished convention should be “radiomic features”
Line 70: A similar approach was proposed
Lines 74-75: Resulting in a better … . In the same line, between over and CT please remove the double spacing.
From this point on I will not highlight all grammar or English errors, just highlighted some to show the authors that English improvements are still required.
Reply: Thank you for your precise corrections, we have modified the text accordingly. In line 57 we have added two references of studies which tested Radiomics in differential diagnosis between lung cancer and granulomas (PMID 32371184 and 33743483). In addition, the text has been revised by Dr. Marwen Eid, a native American English speaker.
Line 140: “in the French and Canadian cohorts”. First time you mention that there are French and Canadian cohorts…
Lines 167-169: You should comment the Table and Figure in the main text. In the Figure caption, please correct “lung cancer”.
Lines 181-184: The list of features provides no value unless it can be contextualized (i.e. compared with the results of other works) or they where validated in other works (i.e. using a different database).
Line 204: “Moving to prognosis, several studies has been performed up to now and some sig nificant correlation has been found between radiomic and outcome data, but not relevant 205 results has been demonstrated in identifying BRCA mutation panel”. Please rephrase.
Line 222: in which Radiomics showing -> No verb… “in which Radiomics showed”?.
Line 244: Figure 2. I do not understand what this Figure contributes to. It seems to be the same Figure than 2 just replacing Ovarian with Prostate cancer.
Line 247: an helpful method. -> a helpful method.
Line 279: Again, “Radiomics signature". According to the established convention, this should be “radiomic signature”
Line 298: “in assessment" -> in the assessment. In addition, shall rephrase as “tumor grading and local invasion, and only few studies tested Radiomics”, I do not think that the next item is part of the same list.
Reply: We want to thank for your precious comments, we have modified the text accordingly. In particular, we have decided to maintain the Figure 2, that could explain in a graphic manner the steps of radiomic signature training and validation. We hope that it will be acceptable.
I wish you all the best,
Sincerely,
Prof Andrea Laghi and co-authors.
Reviewer 2 Report
Thanks for revising the manuscript. I do not have any further comments.
Author Response
We want to thank you for the appreciable evaluation.
Round 3
Reviewer 1 Report
The authors have performed a great job with the manuscript in this second review. They have finally decided to perform an English revision that was very much needed.
Some minor additional considerations:
- Please perform a final spelling check (i.e. Line 32 "perspectives o become -> "perspectives TO become".
- Regarding the consensus about using "Radiomics" capitalized as a name and radiomic as an adjective, please be consistent (See lines 53, 64, and so on).
Author Response
Dear Editor, Dear Reviewers,
We would like to sincerely thank you for your precise review Part II of our manuscript and your overall evaluation.
Please find responses to your comments and consecutive changes based on your recommendations below.
Thank you again.
The authors have performed a great job with the manuscript in this second review. They have finally decided to perform an English revision that was very much needed.
Reply: We want to thank you for your positive evaluation.
Some minor additional considerations:
- Please perform a final spelling check (i.e. Line 32 "perspectives o become -> "perspectives TO become".
- Regarding the consensus about using "Radiomics" capitalized as a name and radiomic as an adjective, please be consistent (See lines 53, 64, and so on).
Reply: Thank you for your precise corrections, we have modified and improved the text accordingly.
I wish you all the best,
Sincerely,
Prof Andrea Laghi and co-authors.